# Lipid Signature of Motile Human Sperm: Characterization of Sphingomyelin, Ceramide, and Phospholipids with a Focus on Very Long Chain Polyunsaturated Fatty Acids

**DOI:** 10.3390/ijms26199301

**Published:** 2025-09-23

**Authors:** Gerardo Martín Oresti, Jessica Mariela Luquez, Silvia Alejandra Belmonte

**Affiliations:** 1Instituto de Investigaciones Bioquímicas de Bahía Blanca (INIBIBB), Consejo Nacional de Investigaciones Científicas y Técnicas (CONICET), Bahía Blanca B8000, Argentina; 2Departamento de Biología, Bioquímica y Farmacia, Universidad Nacional del Sur (UNS), Bahía Blanca B8000, Argentina; 3Instituto de Histología y Embriología de Mendoza (IHEM) “Dr. Mario H. Burgos”, Consejo Nacional de Investigaciones Científicas y Técnicas (CONICET), Universidad Nacional de Cuyo, Centro Universitario, Mendoza M5502JMA, Argentina; 4Facultad de Ciencias Médicas, Universidad Nacional de Cuyo, Mendoza M5502JMA, Argentina

**Keywords:** sphingolipids, ceramide, sphingomyelin, phospholipids, VLPUFA, human sperm

## Abstract

Sperm membrane lipids play a crucial role in male fertility, influencing sperm motility, viability, and functional competence. This study comprehensively characterizes the phospholipid and sphingolipid composition in highly motile human spermatozoa obtained through the swim-up method, a widely used technique in assisted reproductive technology (ART). Using two-dimensional thin-layer chromatography and phosphorus analysis, we identified choline glycerophospholipids (CGP, 45%), ethanolamine glycerophospholipids (EGP, 26%), and sphingomyelin (SM, 17%) as predominant phospholipids, with minor components including cardiolipin, lysophospholipids, phosphatidylinositol, phosphatidylserine, phosphatidic acid, and neutral lipids. Gas chromatography analysis of glycerophospholipids (GPL) revealed a high long chain (C20–C22) polyunsaturated fatty acids (PUFA) content (46.3%), particularly docosahexaenoic acid (DHA, 22:6n-3), which was more abundant in CGP (46%) than EGP (26%). Sphingolipid analysis indicated that ceramide (Cer) and SM shared similar fatty acid profiles due to their metabolic relationship, with very-long-chain (VLC) PUFA (≥C26) being more prevalent in SM (10%) than in Cer (6%). Additionally, argentation chromatography identified highly unsaturated VLCPUFA species in Cer, including 28:3n-6, 28:4n-6, and 30:4n-6, which had not been previously quantified in motile human spermatozoa. Given the essential function of sphingolipid metabolism in spermatogenesis, capacitation, and acrosomal exocytosis, our findings suggest that the balance of VLCPUFA-containing SM and Cer could play a role in sperm performance and fertilization potential. This study provides novel insights into the lipid signature of human sperm and highlights the relevance of membrane lipid remodeling for male fertility and ART outcomes.

## 1. Introduction

The importance of the lipid composition of the sperm plasma membrane lies in its unique characteristics, which are closely related to the ability of these fully differentiated cells to fertilize the oocyte successfully [1]. In both murine and large mammal spermatozoa, the high content of C20–C22 polyunsaturated fatty acids in their glycerophospholipids (GPL) [2,3,4] and the presence of sphingolipids (SLs) with very-long-chain polyunsaturated fatty acids ≥C26 (VLCPUFA) stand out [5,6]. It is worth noting that these VLCPUFA are exclusively bound to sperm sphingomyelins (SM) and not to other phosphoglycerolipids [5]. Notably, SM with VLCPUFA is solely located in the heads of bull and rat spermatozoa [6,7]. The VLCPUFA are produced through the elongation and desaturation of n-3 and n-6 dietary C18-C22 PUFA by a series of elongases of very long chain fatty acids (ELOVL) and desaturases enzymes, involving the sequential action of ELOVL2, ELOVL5, and especially ELOVL4. The latter is expressed in a limited number of tissues, including the skin, brain, retina, Meibomian glands, and testes [8,9].

In the rat testicular environment, Santiago Valtierra et al. (2018) [10] demonstrated that the expression and activity of ELOVL4 correlate with the abundance of VLCPUFA. These unusual molecular species are incorporated into SLs compounds through the enzymatic action of ceramide synthase 3 (CerS3) in mice testis [11]. When *CerS3* is conditionally deleted, it results in an almost complete absence of SL products harboring VLCPUFA, increased apoptosis during meiosis, and spermatogenic arrest that culminates in infertility [12].

The requirement of VLCPUFA in human fertility is underscored by observations correlating diminished levels of these fatty acids in sperm with reduced quantity and quality of spermatozoa, as reported by Craig et al. in 2019 [13]. Although the majority of these investigations have focused on the total semen samples from various mammalian species, including humans, our study aims to elucidate the fatty acid composition of main GPL, SM, and Cer within the membranes of human sperm isolated by swim up commonly used for in vitro fertilization treatment. This issue has not yet been explored. Our findings have prompted us to investigate the relationship between lipids and the human sperm functions we previously studied. Maintaining lipid homeostasis, crucial for cell health, significantly impacts sperm viability, development, and performance during fertilization.

## 2. Results

### 2.1. Phospholipid Composition and Fatty Acid of Total Glycerophospholipids (GPL) in Non-Capacitated Motile Human Sperm

To assess the qualitative and quantitative phospholipid profile of motile non-capacitated human sperm we first resolved the lipids in a two-dimensional TLC (Figure 1a). After the exposure of the plates to iodine vapors, we transferred the silica gel containing each lipid class to different tubes. Then, we quantified the phospholipids by phosphorus analysis. The results showed in Figure 1a,b revealed the lipid composition of sperm membranes, which are choline glycerophospholipids (CGP), diphosphatidylglycerol, commonly known as “cardiolipin” (DPG), ethanolamine glycerophospholipids (EGP), free fatty acids (FFA), lysophosphatidylcholine (LPC), lysophosphoethanolamine (LPE), phosphatidylinositol (PI), phosphatidylserine (PS), sphingomyelin (SM), phosphatidic acid (PA), seminolipid (SL), neutral lipids (NL, e.g., cholesterol and their esters) and some unknown lipids indicated with a question mark.

As expected, we found that major PL were CGP, EGP, and SM that accounted on average 45%, 26%, and 17%, respectively, and minor percentages (3% or less) of lysoPLs (LPC and LPE), PI, PS, DPG followed by negligible amounts of PA (Figure 1c). The CGP/EGP and CGP/SM were 1.73 and 2.64, respectively.

Then, aliquots of isolated total PL from human sperm membranes were analyzed by GC to determine the fatty acid composition of the GPL. As shown in Figure 1c, saturated FA represented 37.9 ± 1.5%, with 16:0 and 18:0 being the most prevalent in human sperm samples. Further, we found MFA (9.5% ± 1.1, mainly 18:1) and DFA (5.9% ± 0.4, mainly 18:2). Strikingly, PUFA constituted 46.3% of the total sperm GPL fatty acids and, as observed in Figure 1c, 22:6n-3 is the most abundant PUFA (32%), followed by minor amounts of 20:3n-6 and 20:4n-6. In addition, we found that only 0.5% of the total FA are C24-26 VLCPUFA in human sperm GPL. This fatty acid composition pattern, with some differences, was similar in major GPL, CGP, and EGP. Notably, the percentage of 22:6n-3 in CGP reached 46%, while in EGP it was 26% (Figure 1c). The latter was characterized by high amounts of molecular species of 16:0, about 36% of total fatty acids (Figure 1c).

### 2.2. Fatty Acid Composition in Sphingomyelin (SM) and Ceramide (Cer) from Human Motile Spermatozoa

Our investigation was particularly directed towards the fatty acid composition of the SM, and especially Cer molecular species isolated from the membranes of human motile spermatozoa obtained via swim up. In Figure 2, representative chromatograms of Cer and SM are shown, demonstrating that both lipids had the same fatty acids, which is expected due to their metabolic relationship. The total content of SM per unit of lipid phosphorus is 6.5 times greater than that of Cer (Figure 2b), and the latter exhibited the same molecular species of VLCPUFAs with ≥C26 carbon atoms as SM. Furthermore, when comparing the proportions of fatty acid groups, no differences were observed between the two lipids, with saturated (S), monoenoic (M), and dienoic (D) fatty acids, ranging from 56 to 58%, 25 to 30%, and 6 to 8%, respectively. However, VLCPUFAs (V) were proportionally more abundant in SM than in Cer, at 10% compared to 6%, respectively (*p* < 0.05).

To concentrate individual molecular species of ≥C26 VLCPUFAs from the minor Cer, we resolved the fatty acid methyl esters (FAME) using argentation TLC. This method allowed us to separate total FAME into discrete bands containing saturated, monoenoic, and dienoic fatty acids, as well as those containing trienoic, tetraenoic, and pentaenoic VLCPUFA series. Each molecular species was then identified by GC (Figure 2c). Notably, 28:3n-6, 28:4n-6, and 30:4n-6 were the most abundant VLCPUFAs found in human sperm Cer (Figure 2c and Figure 3). Additionally, the longest VLCPUFA detected (Figure 2c, lower panel) and quantified (Figure 3) was 32:4n-6.

In the quantitative comparison of fatty acid profiles of Cer and SM in human male gametes, we observed a predominance of saturated and monounsaturated fatty acids, ranging from 16 to 26 carbon atoms (Figure 2c and Figure 3). Notably, palmitic acid (16:0) was the most abundant, accounting for 18.0% of Cer and 32.5% of SM fatty acids. As shown in Figure 3, the proportion of 16:0 in SM is nearly double that of Cer. In contrast, Cer exhibited approximately double the proportions of the longer fatty acids 22:0 and 24:0 compared to SM, with 9.5% vs. 4.7% and 13.1% vs. 5.4%, respectively.

Within the individual molecular species of ≥C26 VLCPUFAs, no qualitative differences were found, but quantitative differences were observed. In both lipids, the predominant molecular species of ceramides (Cer) and sphingomyelins (SM) were 28:3n-6, 28:4n-6, and 30:4n-6, although the proportions were slightly different: 4.4/1.6/2.3 in SM, while they were 2.5/1.0/1.4 in Cer (Figure 3).

## 3. Discussion

This study aimed to elucidate the comprehensive lipid composition of human sperm membranes of motile spermatozoa before the capacitation. Maintaining lipid homeostasis is crucial for sperm health, influencing the survival and functionality of male gametes during fertilization [14]. Thus, we characterized membrane lipids in human spermatozoa with high motility obtained by swim up that are commonly used for in vitro fertilization assays, with a special focus on molecular species of GPL and the major sphingolipids SM and Cer.

Here, we showed that, alongside the abundant GPL rich in 22:6n-3 and the “common” SM and Cer with saturated and monounsaturated fatty acids, molecular species of SM and Cer incorporating VLCPUFA are quantitatively critical constituents of human sperm membranes. In addition, this is the first report showing the quantification of endogenous molecular species of Cer pool in motile human sperm cells obtained by the same method used in assisted reproductive technology procedures. Interestingly, we found that SM from motile spermatozoa had a higher percentage of VLCPUFA (about 10% of total fatty acid) in comparison with previous reports on total sperm cells, where VLCPUFA containing SM comprised 0 to 6.1% of the overall SM pool (mean 2.1%) [13,15]. This is likely attributable to our analysis being conducted on a selected sperm population with the highest motility. Our findings stress the importance of the VLCPUFA concentration in SM for sperm functionality.

This prospect prompts intriguing inquiries into the characteristics and significance of VLCPUFA within SL in human sperm membranes. Recognizing the presence of significant amounts of SM and Cer containing “unusual” fatty acids in spermatozoa could facilitate exploration of their role in male reproductive physiology, potentially serving as key players in fertilization-related processes. Indeed, Craig and colleagues [13] investigated this issue in the human sperm total cell pool, noting that lower levels of VLCPUFA containing SM were strongly associated with reduced sperm count and total motility.

Sphingoglycolipids and the sphingophospholipid, SM, constitute the main SLs within eukaryotic cells, playing crucial roles in shaping membrane structure and functionality [16]. Over the past decade, there has been a lot of interest in these lipids due to certain metabolites they generate, such as sphingoid bases, phosphorylated sphingoid bases, and Cer, which are now essential messengers in cell signaling [16,17,18,19,20,21]. SM and Cer alterations in their molar ratio impact membrane physical properties, influencing microdomain formation, vesicular trafficking, membrane fusion, as well as different processes involving membrane dynamics [22,23]. Specific sphingomyelinases catalyze the hydrolysis of human sperm SM [21] generating most of the ceramide present in the male gamete. In rodents and boar sperm, the enzyme activity rises just before fertilization hydrolyzing SM, to produce ceramides during the acrosome reaction, with a consistent SM drop [24,25,26,27]. Consequently, the steady-state levels of Cer in cells can be modulated by a set of enzymes, which remove or modify the SM or Cer. Cer is a minor lipid class in most animal tissues and cells but is essential for human sperm function and fertilizing capability [19,21,28,29]. Sphingomyelin (SM) molecules are known to associate with cholesterol in membrane microdomains, or lipid rafts, which serve as platforms for protein attachment and signal transduction (for a review see [29]). However, compared to conventional SM species such as 16:0 SM, very long-chain polyunsaturated fatty acid (VLCPUFA)-containing SMs display atypical biophysical properties, including lower transition temperatures and a reduced tendency to form laterally segregated domains with cholesterol in giant unilamellar vesicles [30]. These characteristics suggest that such SM species are unlikely to be incorporated into classical cholesterol-rich membrane rafts in spermatogenic cells. Supporting this, studies on native membranes from differentiating spermatogenic cells have shown that SM and ceramide species containing VLCPUFAs are excluded from the low-density, raft-like light membrane fractions [31].

We hypothesize that the fatty acid composition of these SLs plays a crucial role in their biological function in membrane fusion. In secretory cells, SM is primarily regarded as a structural lipid, yet it can also generate metabolites with highly bioactive properties like Cer, ceramide 1-phosphate (C1P), or sphingosine 1-phosphate (S1P) [32,33]. Furthermore, it remains unclear whether the incorporation of a VLCPUFA into the molecular structure of the distinctive sperm ceramides affects their bioactivity, and, if so, in what manner.

Some authors have demonstrated the accumulation of VLCPUFA-rich SM, Cer, and, even glycosphingolipids (e.g., fucosylated GSL) in normal spermatogenic cells across different species [6,11,12,34,35,36]. This finding underscores the incorporation of these sphingolipids into the spermatozoa where they play crucial physiological roles. The importance of different sphingolipids in human male reproductive physiology has been demonstrated. Our laboratory has elucidated the signaling cascades mediated by different sphingolipids during sperm acrosomal exocytosis [18,19,21]. Some enzymes involved in SL metabolism, such as neutral-sphingomyelinase, ceramidases, ceramide synthase [21], sphingosine kinase 1 [18], and ceramide kinase [19], are present in fully differentiated and terminal cells like the human sperm. We determined how some of these enzymes and SL regulate exocytosis. Cer induces the acrosome reaction and enhances the gamete response to progesterone. Furthermore, we outlined the signaling sequence linking ceramide to the internal and external mobilization of calcium during acrosome secretion [21], demonstrating that some enzymes involved in sphingolipid metabolism are active. For example, the ceramide kinase is able to synthesize C1P from sperm ceramide under a calcium increase during spermatozoa physiological function [19]. In addition, the ceramide effect is mainly due to C1P synthesis. On the other hand, progesterone needs ceramide kinase activity to rise intracellular calcium and induce the acrosome release. S1P, a bioactive sphingolipid, initiates the acrosome reaction by binding to Gi-coupled receptors, thereby activating extracellular calcium influx and calcium efflux from intracellular reserves [18].

Numerous studies in sperm and the testis of different animal species stress the importance of VLCPUFA in the reproductive physiology. The C20–C24 PUFA and ≥C26 VLCPUFA in spermatogenic cells originate from fatty acids derived from the diet, including those from the n-6 series derived from linoleic acid, and the n-3 series derived from linolenic acid (Figure 4). The PUFA elongases ELOVL5 and ELOVL2 are recognized for synthesizing C18–C22 and C20–C24 PUFAs of the n-3 and n-6 series, respectively, operating both sequentially and in conjunction with position-specific fatty acid desaturases. These enzymes are essential for the major PUFAs (C20 and C22) found in the glycerophospholipids of mammalian cells [37,38]. Previously, it has been demonstrated that lysophosphatidic acid acyltransferase 3 (LPAAT3) is essential for incorporating 22:6n-3 into the membrane GPLs of differentiating germ cells. In LPAAT3-KO mice, there is a drastic and specific decrease in DHA-containing phospholipids, leading to male infertility due to a failure in spermiogenesis [39]. In addition, PUFA-rich GPLs, especially those containing DHA, facilitate membrane reshaping and flexibility [40], a crucial characteristic for spermatogenic cells to progress through spermiogenesis and generate spermatozoa. The mature gamete retains these GPL molecular species until their final destination in the oviduct. This suggests that maintaining this membrane deformability is important even up to the moment of fertilization.

Elongases of very long chain fatty acids can elongate C20-C24 PUFAs up to C32 VLCPUFA, a process mediated by the elongase ELOVL4 [10], in rat seminiferous tubules [41] and, more recently, in isolated germ cells [10]. The critical role of ELOVL2 in PUFA and VLCPUFA synthesis in germ cell lipids is highlighted by the fact that *Elovl2*−/− mice are sterile, with their testes containing only spermatogonia and primary spermatocytes [42].

The production of sphingolipids with VLCPUFA in spermatogenic cells relies on the *Cers3* gene [11]. When *Cers3* is specifically deleted in the germ cells of mice, leading to the lack of VLCPUFA-containing Cer, SM, GlcCer, and complex GSL, it causes a halt in spermatogenesis due to enhanced apoptosis during meiosis and formation of multinuclear giant cells [12]. Consequently, in rodents, both simple and complex sphingolipids containing VLCPUFA are crucial for normal spermatogenesis and male fertility. Interestingly, *CERS3*-mRNA and SLs with VLCPUFA were virtually absent in infertile human patients with Sertoli cell-only syndrome, and normal adult human testes contain the same molecular species of VLCPUFA-containing Cer and SM that we observed in motile spermatozoa [12]. This demonstrates that these lipids, along with VLCPUFAs themselves, play an early role in testis physiology and are preserved during epididymal transit, contributing to the function of fully differentiated, fertilization-ready spermatozoa.

Conversely, changes in the SM/Cer ratio (including those molecular species with VLCPUFA) induce changes in rat sperm membrane stability [43]. Our results show that human sperm Cer share the same fatty acids as sphingomyelins (SM), suggesting that Cer may originate from the action of sphingomyelinase on pre-existing SM. In systems containing SM where the Cer ratio is progressively increased by sphingomyelinase activity, the formation of Cer leads to changes in lipid bilayer properties. Cer molecules spontaneously associate to form Cer-enriched microdomains that fuse into large Cer-rich membrane platforms as Cer concentration increases [44]. These Cer structures segregate from the bulk liquid-crystalline fluid phase and exhibit gel-like properties. These properties depend not only on the amount but also the acyl chain length and unsaturation of the generated Cer [45].

Ceramides are characterized by an inverted cone molecular shape, which generally promotes the formation of negatively curved membrane structures—an essential feature for membrane fusion events such as the acrosome reaction [21]. During the acrosome exocytosis, the outer acrosomal membrane bends to make contact with the overlying plasma membrane, facilitating membrane apposition and subsequent fusion. These multiple contact points lead to the release of hydrolytic enzymes from the acrosome, which are believed to be necessary for penetration of the oocyte’s zona pellucida. The geometry and physical behavior of ceramides are likely to reduce the energy barrier necessary for such fusion processes.

Moreover, thermodynamic studies have shown that VLCPUFA-Cer undergo phase transitions close to physiological temperatures when organized as bulk dispersions [30], suggesting a critical role in modulating membrane fluidity and stability. Supporting this, it has shown a significant reduction in the energy required to deform lipid bilayers in the presence of VLCPUFA-Cer [24]. This indicates that VLCPUFA-Cer increases bilayer instability more effectively, potentially favoring local curvature stress and membrane fusion.

These observations are consistent with the idea that the long and highly unsaturated acyl chains of VLCPUFA-Cer may enhance the fusogenic properties of membranes more than those induced by more common saturated and monoenoic ceramide species. Additionally, it has been shown that VLCPUFA-Cer significantly lowers the energy required to deform lipid bilayers [24], indicating that these molecules may enhance bilayer instability, thereby facilitating the generation of local curvature stress and promoting membrane fusion.

Thus, the enrichment of VLCPUFA in sperm SLs may not only serve a structural role but also act as a functional determinant in enabling the membrane remodeling events essential for successful fertilization. Consistent with the varying levels of Cer containing VLCPUFA, changes in cell membrane biophysical properties [24,43] may further enhance, along with other enzymatic and signaling processes, the development of motility and/or the acrosome reaction (Figure 4).

In summary, the constituents of sperm membranes play a crucial role in development and sperm function. Recent lipidomics studies have highlighted the importance of membrane lipids—such as sulfogalactosylglycerolipid (SGG, seminolipid), cholesterol sulfate, and GPL with PUFAs—as key predictors of semen quality [46]. Notably, SLs with VLCPUFA have emerged as particularly significant.

## 4. Materials and Methods

### 4.1. Ethics Statement and Human Sperm Preparation

Twelve healthy male donors provided semen samples (age range: 18 to 40 years). Donors were excluded if they were unable to provide a sample, had a current or recent history of sexually transmitted infections, had undergone a vasectomy, or had a known testicular obstruction. In all cases, semen samples were collected via masturbation following a minimum of 48 h of sexual abstinence. We used only semen samples that accomplished the World Health Organization (WHO, 2021) [47] specifications, for the experiments shown here. Data collection adheres to the guidelines established in Argentina (ANMAT 5330/97) and the International Declaration of Helsinki. All donors signed an informed consent form according to supply semen samples. The protocol for semen manipulation was accepted by the Ethics Committee of the School of Medicine, National University of Cuyo. After semen liquefaction (30–60 min at 37 °C), highly motile sperm were recovered after a swim-up separation for 1 h in HTF (5.94 g/liter NaCl, 0.35 g/liter KCl, 0.05 g/liter MgSO_4_·7H_2_O, 0.05 g/liter KH_2_PO_4_, 0.3 g/liter CaCl_2_·2H_2_O, 2.1 g/liter NaHCO_3_, 0.51 g/liter glucose, 0.036 g/liter sodium pyruvate, 2.39 g/liter sodium lactate, 0.06 g/liter penicillin, 0.05 g/liter streptomycin, 0.01 g/liter phenol red supplemented with 5 mg/mL of BSA) at 37 °C in an atmosphere of 5% CO_2_, 95% air. Cell concentration was then adjusted with HTF to 10 × 10^6^ sperm/mL. We developed the protocol as described in our publications [18,19,21,48,49,50].

### 4.2. Lipid Separation and Analysis

The bulk of sperm suspensions obtained were then subjected to a gentle centrifugation (5 min at 400× *g*). After collecting cells, lipid extracts were prepared and partitioned according to Bligh and Dyer [51]. Aliquots from these extracts were taken to determine the total phospholipid (PL) content in the samples by measuring the amount of lipid phosphorus [52].

For preparative isolation of lipid classes, most of the lipid extracts were spotted on TLC plates (500 µm, silica gel G) under N_2_, along with commercial standards (Sigma Chemical Co., St. Louis, MO, USA). The polar lipids remained at the origin of the plates, and the neutral lipids were resolved in two steps. Chloroform: methanol: aqueous ammonia (90:10:2 by vol.) was run up to the middle of the plates to separate the ceramides. Then, these solvents were evaporated and the plates were developed again by running n-hexane:diethyl ether (80:20, by vol.) up to the top of the plates to resolve minor neutral lipids (e.g., cholesterol esters and/or triacylglycerols).

The total phospholipid fraction was subjected to further separations and analyses. Aliquots were set aside to study the total GPL fatty acid composition. Most of the rest was used for preparative isolation of SM and major GPL using chloroform: methanol:acetic acid:0.15 mol/L NaCl (50:25:8:2.5, by vol.). The phospholipids were resolved into classes by two-dimensional TLC [53]. The spots containing each phospholipid class were scraped from the plates followed by elution and quantification by phosphorus analysis in eluate aliquots. After drying, other aliquots of choline glycerophospholipids (CGP) and ethanolamine glycerophospholipids (EGP) were subjected to study fatty acid composition.

A mild alkali treatment was performed on the SM and Cer samples in order to remove any potential lipid contaminant containing ester-bound fatty acids. Both lipids were taken to dryness and treated (under N_2_) with 0.5 N NaOH in anhydrous methanol at 50 °C for 10 min. After this alkaline treatment, SM and Cer were recovered again by TLC. This procedure, involving alkaline methanolysis, was also used to obtain, as methyl esters, the fatty acids ester-bound to total GPL (thus excluding the fatty acids amide-bound to SM from this group). All solvents used in this study were HPLC-grade (JT Baker, Phillipsburg, NJ, USA; UVE, Dorwill, Argentina), and most procedures were carried out under N_2_.

With the exception of the data shown in Figure 1, in which lipid classes were located with iodine vapors, lipids were located under UV light after spraying the plates with dichlorofluorescein, scraped from the plates and collected into tubes for elution. This was achieved by three successive extractions of the silica support by vigorously mixing it with chloroform:methanol:water (5:5:1, by vol.), centrifuging, collecting the solvents, and partitioning the resulting solvent mixtures with four volumes of water to recover the lipids in the organic phases.

After elution, a fatty acid analysis of Cer and SM was performed after converting the lipids to fatty acid methyl esters (FAME), followed by gas-chromatography (GC), using the conditions and instrumentation described in previous work [31]. Briefly, after adding appropriate internal standards, transesterification was performed with 0.5N H_2_SO_4_ in N_2_-saturated anhydrous methanol by keeping the samples overnight at 45 °C under N_2_ in screw-capped tubes. The resulting FAME were routinely purified by TLC on pre-cleaned silica Gel G plates using hexane:ether (95:5, by vol.) and then injected in the GC instrument.

To resolve according unsaturation, aliquots of FAME from Cer were subjected to argentation thin-layer chromatography (TLC) (Silica Gel G:AgNO_3_, 80:20 by weight, and chloroform:methanol, 90:10 by vol.), which separated the esters in different bands containing saturates and different unsaturated fractions. After elution, these fractions were analyzed by GC/FID to identify the VLCPUFA linked to human sperm Cer.

### 4.3. Statistical Analysis

Data are presented as mean values ± SD from at least three sample pools, each obtained from independent cell preparations. Statistical analysis was performed using the Graph Pad Prism software, version 5.0 © (San Diego, CA, USA). The Student *t*-test was used to compare differences between two datasets. *p* values < 0.05 were considered significant.

## Figures and Tables

**Figure 1 ijms-26-09301-f001:**
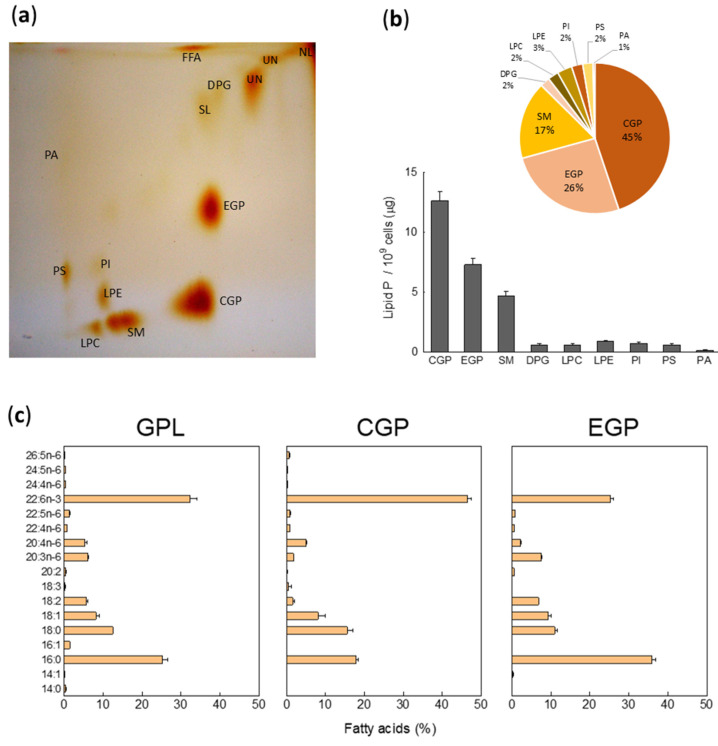
Qualitative phospholipid and fatty acid profile of human spermatozoa obtained by swim-up. (**a**) Shows a representative TLC to resolve phospholipids. To perform this two-dimensional TLC, two solvents were used, chloroform/methanol/ammonia (65:25:5) and chloroform/acetone/methanol/acetic acid/water (30:40:10:10:5 by vol.). The lipids were located by exposure of the plates to iodine vapors. (**b**) Phospholipid composition (upper panel) and amounts per cell estimated from lipid phosphorus (bottom panel) of each phospholipid constituents of human spermatozoa. (**c**) Fatty acid composition of total glycerophospholipids (GPL), choline and ethanolamine glycerophospholipids (CGP and EGP, respectively). Other abbreviations: DPG, diphosphatidylglycerol (cardiolipin); FFA, free fatty acids; LPC, lysophosphatidylcholine; LPE, lysophosphoethanolamine; PI, phosphatidylinositol; PS, phosphatidylserine; SM, sphingomyelin; PA, phosphatidic acid; SL, Seminolipid; NL, neutral lipids; UN, unknown lipids.

**Figure 2 ijms-26-09301-f002:**
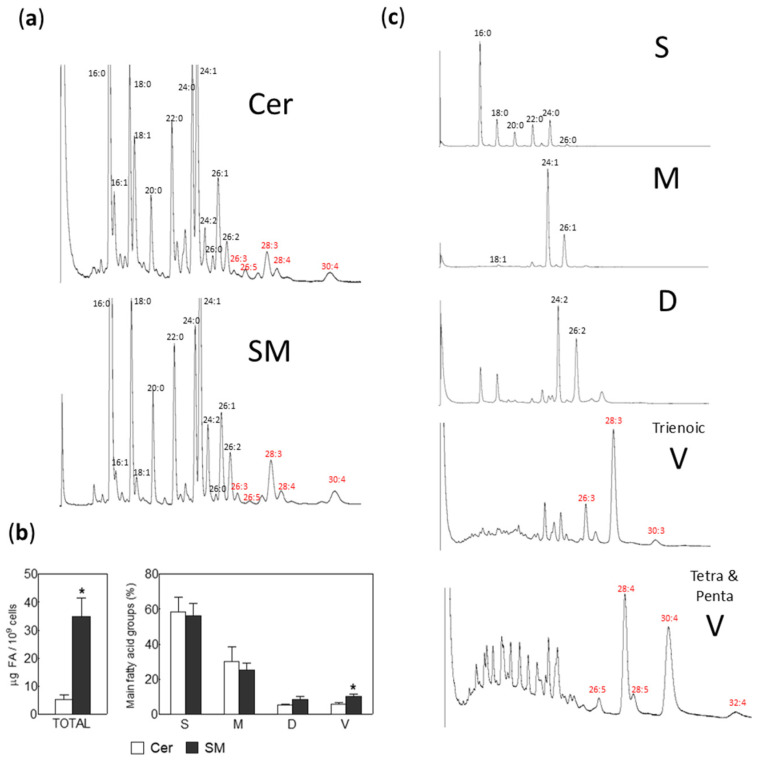
Ceramide (Cer) and sphingomyelin (SM) fatty acid profile of human spermatozoa obtained by swim up. (**a**) Shows a representative chromatogram displaying the qualitative presence of ≥C26 VLCPUFA. (**b**) Content of Cer and SM. To compare them, the lipids were quantified by their fatty acids and expressed as µg per cell. Left panel: amounts of total fatty acids; right panel, percentages of main fatty acids, grouped into saturated (S), monoenoic (M), dienoic (D), and total VLCPUFA (V) (* *p* < 0.01). (**c**) Enrichment of different groups of fatty acids from sperm Cer, each chromatogram represents a different fraction of fatty acids separated using AgNO_3_-TLC, VLCPUFA fractions were composed by tri-, tetra-, and pentaenoic fatty acids.

**Figure 3 ijms-26-09301-f003:**
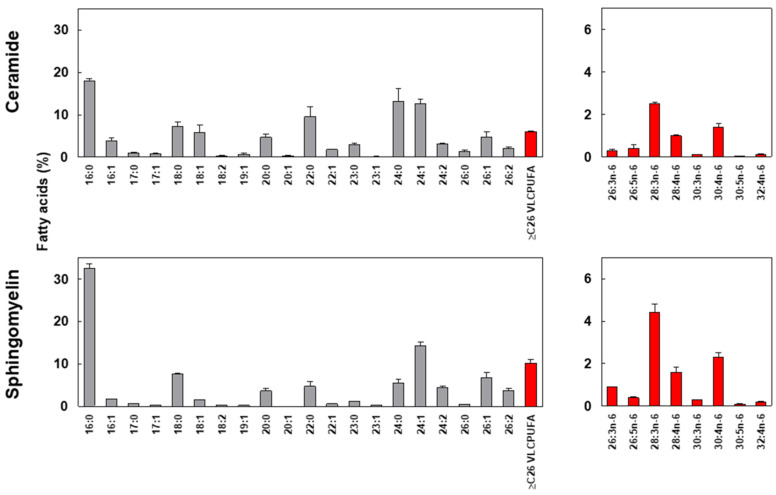
Main individual fatty acids in Cer and SM. Left panels: percentages of individual saturated, monoenoic, and dienoic fatty acids relative to total ≥C26 VLCPUFA. Right panels: percentages of individual ≥C26 VLCPUFA are shown. Notably, in both Cer and SM from human sperm, 28:3n-6, 28:4n-6, and 30:4n-6 are the main VLCPUFA molecular species.

**Figure 4 ijms-26-09301-f004:**
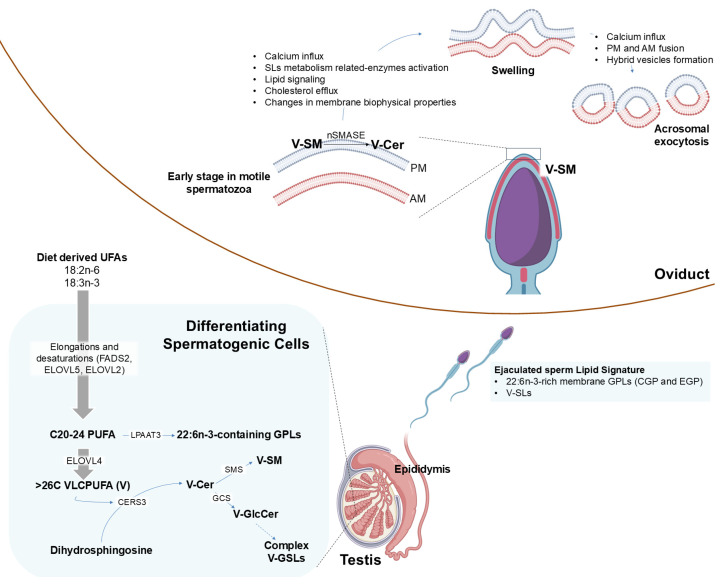
The scheme summarizes the biosynthetic pathways of long (C20–C22) and very long-chain (≥C26) polyunsaturated fatty acids (PUFA and VLCPUFA, respectively) and their incorporation into differentiating spermatogenic cell GPLs and SLs in the testis. After leaving the testis and passing through epididymal transit, ejaculated spermatozoa experience lipid remodeling but conserve 22:6n-3-rich GPLs (CGP and EGP) and VLCPUFA-containing SM. Upon ejaculation, sperm cells carry these molecular species of lipids into the female reproductive tract, where additional lipid remodeling occurs in the oviduct, including the V-SM → V-Cer reaction. These final modifications facilitate membrane fluidity and the ability of spermatozoa to undergo capacitation and the acrosome reaction, both of which are essential for fertilization. Abbreviations: AM, acrosomal membrane; Cer, ceramide; CERS3, ceramide synthase 3; CGP and EGP, choline and ethanolamine glycerophospholipids, respectively; ELOVL, elongation of very-long-chain fatty acid protein; FADS2, Δ6 desaturase; GPL, glycerophospholipids; GlcCer, glucosylceramide; GCS, GlcCer synthase; GSLs, glycosphingolipids; LPAAT3, lysophosphatidic acid acyltransferase 3; nSMASE, neutral sphingomyelinase; PM, plasma membrane; SL, sphingolipids; SMS, SM synthase; SM, sphingomyelin; UFAs, unsaturated fatty acids; V, VLCPUFA. The scheme was created with BioRender.com.

## Data Availability

All relevant data are contained within the manuscript.

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
