# Peer review of "Lipid Signature of Motile Human Sperm: Characterization of Sphingomyelin, Ceramide, and Phospholipids with a Focus on Very Long Chain Polyunsaturated Fatty Acids"

_ijms, 2025, doi:10.3390/ijms26199301_

Round 1

Reviewer 1 Report

Comments and Suggestions for Authors

The manuscript describes for the first time the lipid composition of human sperm collected by the swim-up method, widely used in assisted reproductive technology. The authors focus on sphingomyelin and ceramides containing very long chain (>C26) and polyunsaturated (3-5 double bonds) fatty acids (VLC-PUFAs), which had been described previously to be essential for spermatogenesis and correlate with sperm motility and sperm count. By that the authors provide for the first time data about the detailed VLC-PUFA composition in such sperm, which fits to data obtained from human testes. Furthermore, their results indicate a higher content of SLs with these VLC-PUFAs in sperm collected by the swim-up method than previously reported by total sperm. By that they highlight that sphingolipids with VLC-PUFAs are indeed a marker correlating good quality of sperm.

The data are of interest for reproductive medicine and corresponding basic research and I support publication.

Minor comments.

The impact of the data would have been stronger, if the authors would have had a corresponding total sperm group to compare, rather than to compare with data from the literature.

Fig. 2: The authors annotate a peak of tetra. & penta-unsaturated FAME with 26:5. Is this correct? Or might this be 26:4(n-3)? The result has implications on the substrate specificity of human CerS3. With 26:5(n-3), CerS3 would be able to accept Acyl-CoAs with a double bond in delta-11 position, which might be worth to mention in the discussion.

Discussion line 302: This chapter starts not logically. The previous chapter ends with a discussion of LPAAT3. This chapter shifts to fatty acid elongation without introducing the einzyme group by name. I guess the authors wanted to discuss now the group of ELOVL enzymes. If so, they may start with "Elongasese of very long chain fatty acids (ELOVL) can elongate..." By the way, ELOVL4 is acting also in human skin and produces even C36-fatty acids.

Author Response

REVIEWER 1

Thank you very much for taking the time to review this manuscript. Please find the detailed responses below and the corresponding revisions/changes highlighted in yellow in the re-submitted manuscript file. Please find the reviewed manuscript file attached.

The manuscript has been considerably revised and several changes have been included. We have tried to answer all the criticisms and introduced most of the changes suggested. A point-by-point answer to the comments is included below.

Comment 1:

The manuscript describes for the first time the lipid composition of human sperm collected by the swim-up method, widely used in assisted reproductive technology. The authors focus on sphingomyelin and ceramides containing very long chain (>C26) and polyunsaturated (3-5 double bonds) fatty acids (VLC-PUFAs), which had been described previously to be essential for spermatogenesis and correlate with sperm motility and sperm count. By that, the authors provide for the first time data about the detailed VLC-PUFA composition in such sperm, which fits to data obtained from human testes. Furthermore, their results indicate a higher content of SLs with these VLC-PUFAs in sperm collected by the swim-up method than previously reported by total sperm. By that, they highlight that sphingolipids with VLC-PUFAs are indeed a marker correlating good quality of sperm.

The data are of interest for reproductive medicine and corresponding basic research and I support publication.

Response 1: We sincerely thank you for your positive feedback and for recognizing the novelty and relevance of our work. We are pleased you consider our findings valuable for reproductive medicine and basic research. We have carefully considered and addressed the minor issues you raised, as detailed in our responses to the following comments.

Minor comments

Comment 2:

The impact of the data would have been stronger, if the authors would have had a corresponding total sperm group to compare, rather than to compare with data from the literature.

Response 2:

Thank you for your valuable suggestion. We fully agree that including a corresponding total sperm group as a direct comparator would have added further weight to our findings. However, the design of our study was purposefully focused on analyzing the lipid profile of highly motile spermatozoa obtained via the swim-up method, a standard procedure in assisted reproductive technology (ART). This focus aimed to better understand the molecular composition of sperm subpopulations with proven fertilizing potential.

While a direct comparison with total spermatozoa was not feasible within the scope of this study—due to sample limitations and the need for sufficient cell numbers—we carefully selected previously published studies as benchmarks. Specifically, we referenced two robust and peer-reviewed datasets (Craig et al., 2019, Gavrizi et al., 2023). The former involved a well-characterized cohort with a large sample size (n=70) spanning a broad age range (18–55 years). The latter linked VLCPUFA levels in sperm to semen quality and reproductive outcomes across a multicenter trial. These studies are statistically sound, methodologically aligned, and highly relevant, allowing for the contextualization of our results within a broader biological framework.

Our findings go beyond the lipid quantification by identifying an enrichment of VLCPUFA in sphingomyelin from motile sperm cells—levels notably higher than those previously reported for total sperm populations. This suggests a functional specialization in the lipid composition of motile spermatozoa that may be obscured in total semen analyses.

The rationale has been clarified in the Discussion section, lines 228-234, explicitly acknowledging the absence of a matched total sperm group as a limitation while highlighting the justification for our comparative approach and the relevance of our findings for ART-focused research.

Comment 3:

Fig. 2: The authors annotate a peak of tetra & penta-unsaturated FAME with 26:5. Is this correct? Or might this be 26:4(n-3)? The result has implications for the substrate specificity of human CerS3. With 26:5(n-3), CerS3 would be able to accept Acyl-CoAs with a double bond in the delta-11 position, which might be worth to mention in the discussion.

Response 3: Thank you for pointing this out. After carefully reviewing the data, we confirmed that the annotation is accurate and should refer to 26:5(n-6) rather than 26:4(n-3). Based on the retention time, we found that 26:5(n-6) is a minor component among the VLCPUFAs present in sperm ceramide (Cer) and sphingomyelin (SM). We also acknowledge that without the enrichment of the tetra- and pentaenoic fraction, detecting this component would have been much more challenging. Furthermore, similar annotations have been reported in previous studies, which have established that human sperm does not contain ≥C26 VLCPUFAs of the n-3 series, but rather only n-6 series of VLCPUFAs (Poulos et al., 1987, Robinson et al., 1992)

Comments 4:

Discussion line 302: This chapter starts not logically. The previous chapter ends with a discussion of LPAAT3. This chapter shifts to fatty acid elongation without introducing the enzyme group by name. I guess the authors wanted to discuss now the group of ELOVL enzymes. If so, they may start with "Elongases of very long chain fatty acids (ELOVL) can elongate..." By the way, ELOVL4 is acting also in human skin and produces even C36-fatty acids

Response 4: We agree that the transition between the previous discussion on LPAAT3 and the topic of fatty acid elongation could be improved. Accordingly, we have revised the introduction to this paragraph to introduce the group of ELOVL enzymes and their role in the elongation of polyunsaturated and very long-chain polyunsaturated fatty acids. The revised text now begins with: " Elongases of very long chain fatty acids (ELOVL) can elongate C20-C24 PUFAs up to C32 VLCPUFA, a process mediated by the elongase ELOVL4 (Santiago Valtierra et al., 2018), in rat seminiferous tubules (Aveldano et al., 1993) and, more recently, in isolated germ cells (Santiago Valtierra et al., 2018)." Lines 314-315 (Highlighted in yellow).

Regarding your suggestion to include the role of ELOVL4 in human skin and its capacity to produce C36 fatty acids, we respectfully chose not to incorporate this information in the revised paragraph. We intended to maintain the focus on the elongation of PUFA and VLCPUFA, which is more directly relevant to testicular and sperm lipid metabolism. While ELOVL4 indeed plays a significant role in human skin, its activity there is primarily associated with the elongation of saturated and monoenoic very-long-chain fatty acids (VLCFA), which are characteristic of skin ceramides and not central to the scope of our discussion.

References

AVELDANO, M. I., ROBINSON, B. S., JOHNSON, D. W. & POULOS, A. 1993. Long and very long chain polyunsaturated fatty acids of the n-6 series in rat seminiferous tubules. Active desaturation of 24:4n-6 to 24:5n-6 and concomitant formation of odd and even chain tetraenoic and pentaenoic fatty acids up to C32. J Biol Chem, 268, 11663-9.

CRAIG, L. B., BRUSH, R. S., SULLIVAN, M. T., ZAVY, M. T., AGBAGA, M. P. & ANDERSON, R. E. 2019. Decreased very long chain polyunsaturated fatty acids in sperm correlates with sperm quantity and quality. J Assist Reprod Genet, 36, 1379-1385.

GAVRIZI, S. Z., HOSSEINZADEH, P., BRUSH, R. S., TYTANIC, M., ECKART, E., PECK, J. D., CRAIG, L. B., DIAMOND, M. P., AGBAGA, M. P. & HANSEN, K. R. 2023. Sperm very long-chain polyunsaturated fatty acids: relation to semen parameters and live birth outcome in a multicenter trial. Fertil Steril, 119, 753-760.

POULOS, A., JOHNSON, D. W., BECKMAN, K., WHITE, I. G. & EASTON, C. 1987. Occurrence of unusual molecular species of sphingomyelin containing 28-34-carbon polyenoic fatty acids in ram spermatozoa. Biochem J, 248, 961-4.

ROBINSON, B. S., JOHNSON, D. W. & POULOS, A. 1992. Novel molecular species of sphingomyelin containing 2-hydroxylated polyenoic very-long-chain fatty acids in mammalian testes and spermatozoa. J Biol Chem, 267, 1746-51.

SANTIAGO VALTIERRA, F. X., PENALVA, D. A., LUQUEZ, J. M., FURLAND, N. E., VASQUEZ, C., REYES, J. G., AVELDANO, M. I. & ORESTI, G. M. 2018. Elovl4 and Fa2h expression during rat spermatogenesis: a link to the very-long-chain PUFAs typical of germ cell sphingolipids. J Lipid Res, 59, 1175-1189.

Reviewer 2 Report

Comments and Suggestions for Authors

The lipid composition in the sperm plasma membrane plays a crucial role in sperm function, being closely associated with membrane permeability and fluidity. This manuscript focus on VLCPUFA in ART-relevant sperm populations is novel and well designed. We agree with that this manuscript can be accepted for publication after minor issues are addressed.
1.Authorship & Affiliations: The numbering of affiliations 3 and 4 is inconsistent (both labeled as "3").
2.ABSTRACT: Define abbreviations at first mention. e.g., ART should be spelled out as "assisted reproductive technology"; This sentence "the balance of VLCPUFA-containing SM and Cer emerges as a key factor" can be "the balance of VLCPUFA-containing SM and Cer could play a role……"
3.The hypothesis that VLCPUFA in SLs affects membrane fusion needs more detail discussion. Please cite relevant studies on VLCPUFA’s biophysical effects on membrane rigidity.
4.Materials and methods: please provide more details of donor information (age, fertility status…).

Comments on the Quality of English Language

Pay attention to all abbreviations at first mention.

Author Response

REVIEWER 2

Thank you for reviewing our manuscript. We sincerely appreciate your thoughtful comments and suggestions.

The manuscript has been substantially revised, with several key changes incorporated throughout. We have done our best to address all of your concerns and have implemented most of the suggested revisions. 

A detailed, point-by-point response to each of your comments is provided below. All corresponding changes in the manuscript are highlighted in grey in the re-submitted version. Please find the reviewed manuscript file attached

Comment 1:

The lipid composition in the sperm plasma membrane plays a crucial role in sperm function, being closely associated with membrane permeability and fluidity. This manuscript's focus on VLCPUFA in ART-relevant sperm populations is novel and well-designed. We agree that this manuscript can be accepted for publication after minor issues are addressed.

Response 1: Thank you for your positive feedback and for highlighting the novelty and design of our study. We appreciate your support for the publication of our manuscript. We have carefully addressed the minor issues you raised, as outlined in our responses to the following comments.

Comment 2:

1. Authorship & Affiliations: The numbering of affiliations 3 and 4 is inconsistent (both labeled as "3").

Response 2: Thank you for pointing this out. You are correct — there was an inconsistency in the numbering of affiliations 3 and 4. We have corrected this in the revised manuscript (Highlighted in grey)

 Comment 3:

ABSTRACT: Define abbreviations at first mention. e.g., ART should be spelled out as "assisted reproductive technology"; This sentence "the balance of VLCPUFA-containing SM and Cer emerges as a key factor" can be "the balance of VLCPUFA-containing SM and Cer could play a role……"

Response 3:  ART is defined in line 23, as requested.

We have incorporated the suggested change in line 37, which is highlighted in grey.

Comment 4:

The hypothesis that VLCPUFA in SLs affects membrane fusion needs more detail discussion. Please cite relevant studies on VLCPUFA’s biophysical effects on membrane rigidity.

Response 4: Thank you for this insightful comment. We have expanded the discussion to further support the hypothesis that VLCPUFA-containing sphingolipids (SLs), particularly ceramides (Cer), may influence membrane fusion processes through their unique biophysical properties. We have incorporated these points into the revised discussion and cited the relevant studies that address the biophysical effects of VLCPUFA-Cer on membrane properties. Lines 343-370, highlighted in grey.

…Ceramides are characterized by an inverted cone molecular shape, which generally promotes the formation of negatively curved membrane structures—an essential feature for membrane fusion events such as the acrosome reaction [1]. During the acrosome exocytosis, the outer acrosomal membrane bends to make contact with the overlying plasma membrane, facilitating membrane apposition and subsequent fusion. These multiple contact points lead to the release of hydrolytic enzymes from the acrosome, which are believed to be necessary for penetration of the oocyte's zona pellucida. The geometry and physical behavior of ceramides are likely to reduce the energy barrier necessary for such fusion processes.

Moreover, thermodynamic studies have shown that VLCPUFA-Cer undergoes phase transitions close to physiological temperatures when organized as bulk dispersions [2], suggesting a critical role in modulating membrane fluidity and stability. Supporting this, it has shown a significant reduction in the energy required to deform lipid bilayers in the presence of VLCPUFA-Cer [3]. This indicates that VLCPUFA-Cer increases bilayer instability more effectively, potentially favoring local curvature stress and membrane fusion.

These observations are consistent with the idea that the long and highly unsaturated acyl chains of VLCPUFA-Cer may enhance the fusogenic properties of membranes more than those induced by more common saturated and monoenoic ceramide species. Additionally, it has been shown that VLCPUFA-Cer significantly lowers the energy required to deform lipid bilayers [3], indicating that these molecules may enhance bilayer instability, thereby facilitating the generation of local curvature stress and promoting membrane fusion.

These findings support the notion that the long and highly unsaturated acyl chains of VLCPUFA-Cer enhance the fusogenic properties of membranes more effectively than saturated or monoenoic ceramide species. Thus, the enrichment of VLCPUFA in sperm SLs may not only serve a structural role but also act as a functional determinant in enabling the membrane remodeling events essential for successful fertilization…

 Comment 5:

Materials and methods: please provide more details of donor information (age, fertility status…).

Response 5:  Additional details regarding donor information have been included in the Materials and Methods section (lines 415–419, highlighted in grey).

Twelve healthy male donors provided semen samples (age range: 18 to 40 years). Donors were excluded if they were unable to provide a sample, had a current or recent history of sexually transmitted infections, had undergone a vasectomy, or had a known testicular obstruction. In all cases, semen samples were collected via masturbation following a minimum of 48 hours of sexual abstinence.

References

  1. Vaquer, C.C., et al., Ceramide induces a multicomponent intracellular calcium increase triggering the acrosome secretion in human sperm. Biochim Biophys Acta Mol Cell Res, 2020. 1867(7): p. 118704.
  2. D.A., et al., Atypical surface behavior of ceramides with nonhydroxy and 2-hydroxy very long-chain (C28-C32) PUFAs. Biochim Biophys Acta, 2014. 1838(3): p. 731-8.
  3. Ahumada-Gutierrez, H., et al., Mechanical properties of bilayers containing sperm sphingomyelins and ceramides with very long-chain polyunsaturated fatty acids. Chem Phys Lipids, 2019. 218: p. 178-186.

Reviewer 3 Report

Comments and Suggestions for Authors

The study of Gerardo M. Oresti et al. characterizes the phospholipid and sphingolipid composition and other lipids in highly motile human spermatozoa obtained through the swim-up method, a widely used technique in assisted reproductive technology (ART). This study provides novel insights nto the lipid signature of human sperm and highlights the relevance of membrane lipid remodeling for male fertility and ART outcomes.

Although this study is interesting from a cultural perspective, I would appreciate it if the authors introduced the concept of lipid rafts, which many researchers have described as essential for sperm maturation and reproductive success. The lipids mentioned in this study, particularly cholesterol and glycosphingolipids, are also key components of lipid rafts. Additionally, I would suggest that the authors reference relevant literature on lipid rafts, such as a comprehensive and complete review published in early 2023 discussing the advances and molecular mechanisms involving lipid rafts although in other diseases.

I would appreciate it if the authors also analyzed the lipid composition of sperm from patients with Klinefelter syndrome, as it may differ from that of normozoospermic individuals. Additionally, I would suggest that the authors introduce Klinefelter syndrome and higher-grade X-chromosome aneuploidies, including a citation from 2021 to provide proper scientific context.

Natural compounds could potentially play a role in modulating sperm membrane lipid composition. Various bioactive molecules, such as omega-3 polyunsaturated fatty acids (PUFAs), antioxidants, and phytochemicals, have been shown to influence lipid metabolism, membrane fluidity, and overall sperm function. Given the crucial role of lipids in sperm maturation, motility, and fertilization capacity, it would be valuable for the authors to explore whether dietary or supplemental interventions could impact the lipid profile of sperm. Introducing this concept could provide a broader perspective on possible strategies to optimize male fertility.

Author Response

REVIEWER 3

Thank you very much for taking the time to review our manuscript.

The manuscript has been substantially revised, with some changes incorporated throughout.

A detailed, point-by-point response to each of your comments is provided below. All corresponding changes in the manuscript are highlighted in green in the re-submitted version. Please find the reviewed manuscript file attached.

Comment 1:

The study of Gerardo M. Oresti et al. characterizes the phospholipid and sphingolipid composition and other lipids in highly motile human spermatozoa obtained through the swim-up method, a widely used technique in assisted reproductive technology (ART). This study provides novel insights into the lipid signature of human sperm and highlights the relevance of membrane lipid remodeling for male fertility and ART outcomes.

Response 1: Thank you very much for your positive feedback and for highlighting the significance of our study. We appreciate your support of the manuscript's contribution to the field of reproductive biology.

 Comment 2:

Although this study is interesting from a cultural perspective, I would appreciate it if the authors introduced the concept of lipid rafts, which many researchers have described as essential for sperm maturation and reproductive success. The lipids mentioned in this study, particularly cholesterol and glycosphingolipids, are also key components of lipid rafts. Additionally, I would suggest that the authors reference relevant literature on lipid rafts, such as a comprehensive and complete review published in early 2023 discussing the advances and molecular mechanisms involving lipid rafts although in other diseases.

Response 2.  We appreciate the reviewer’s comment and would like to clarify the relationship between the unique sphingolipid (SL) composition in sperm cells and lipid rafts. To address this point, we have added the following paragraph to the Discussion section (lines 257–268), along with relevant references on the topic. The changes have been highlighted in green.

 Sphingomyelin (SM) molecules are known to associate with cholesterol in membrane microdomains, or lipid rafts, which serve as platforms for protein attachment and signal transduction (For a review see [1]). However, compared to conventional SM species such as 16:0 SM, very long-chain polyunsaturated fatty acid (VLCPUFA)-containing SMs display atypical biophysical properties, including lower transition temperatures and a reduced tendency to form laterally segregated domains with cholesterol in giant unilamellar vesicles [2]. These characteristics suggest that such SM species are unlikely to be incorporated into classical cholesterol-rich membrane rafts in spermatogenic cells. Supporting this, studies on native membranes from differentiating spermatogenic cells have shown that SM and ceramide species containing VLCPUFAs are excluded from the low-density, raft-like light membrane fractions[3].

Comment 3:

I would appreciate it if the authors also analyzed the lipid composition of sperm from patients with Klinefelter syndrome, as it may differ from that of normozoospermic individuals. Additionally, I would suggest that the authors introduce Klinefelter syndrome and higher-grade X-chromosome aneuploidies, including a citation from 2021 to provide proper scientific context.

 Response 3:

We appreciate the reviewer’s interest in the potential lipid differences in sperm from patients with Klinefelter syndrome. However, this topic lies outside the scope of our current study, which specifically investigates the lipid composition of highly motile, normozoospermic spermatozoa obtained through the swim-up method. This methodological focus is directly aligned with clinical practices in assisted reproductive technologies, where only functional, motile sperm cells are utilized.

Klinefelter syndrome is commonly associated with hypergonadotropic hypogonadism, extensive testicular fibrosis, and azoospermia in the vast majority of patients. As a result, spermatozoa are typically absent from the ejaculate, and viable sperm can only be retrieved via invasive testicular extraction techniques (TESE), followed by ICSI in select cases [4, 5]. These clinical realities underscore the fundamental biological and methodological incompatibilities between our study population and individuals with Klinefelter syndrome.

Additionally, analyzing subtle lipid components such as ceramides in these rare testicular sperm cells presents significant technical limitations due to their scarcity and heterogeneity. While one early study [6] reported bulk lipid profiles in Klinefelter ejaculates, the majority of lipids in that context likely originated from accessory gland secretions rather than sperm cells themselves.

Given these constraints, and following the journal’s guidance, we believe that including a discussion on Klinefelter syndrome and related chromosomal aneuploidies—though scientifically interesting—would not enhance the focus or value of the current manuscript. We have therefore opted not to include these references in the revised version.

Comment 4:

Natural compounds could potentially play a role in modulating sperm membrane lipid composition. Various bioactive molecules, such as omega-3 polyunsaturated fatty acids (PUFAs), antioxidants, and phytochemicals, have been shown to influence lipid metabolism, membrane fluidity, and overall sperm function. Given the crucial role of lipids in sperm maturation, motility, and fertilization capacity, it would be valuable for the authors to explore whether dietary or supplemental interventions could impact the lipid profile of sperm. Introducing this concept could provide a broader perspective on possible strategies to optimize male fertility.

Response 4:

We appreciate the reviewer’s thoughtful suggestion and fully agree that natural compounds—particularly omega-3 PUFAs and other bioactive dietary components—may influence sperm lipid metabolism and function. While our study aimed to characterize the endogenous lipid composition of highly motile human sperm obtained via swim-up, we acknowledge the broader relevance of nutritional influences on lipid remodeling and male fertility.

To provide context for this, we have already introduced the concept in the Introduction (Lines 55–57), where we note that “VLCPUFAs are produced through the elongation and desaturation of n-3 and n-6 dietary C18–C22 PUFA by a series of elongase and desaturase enzymes, including ELOVL2, ELOVL5, and especially ELOVL4.” In the Discussion section (Lines 297–302), we also highlight the role of dietary PUFA precursors in shaping membrane properties essential for spermatogenesis and sperm function.

However, investigating the impact of dietary or supplemental interventions on sperm lipid profiles would require a longitudinal, controlled clinical approach involving dietary monitoring or supplementation, which was beyond the scope and experimental design of our current biochemical analysis. Nonetheless, we agree that this is an important direction for future research and appreciate the opportunity to point readers toward this broader perspective.

References

  1. Suhaiman, L. and S.A. Belmonte, Lipid remodeling in acrosome exocytosis: unraveling key players in the human sperm. Front Cell Dev Biol, 2024. 12: p. 1457638.
  2. Penalva, D.A., et al., Atypical surface behavior of ceramides with nonhydroxy and 2-hydroxy very long-chain (C28-C32) PUFAs. Biochim Biophys Acta, 2014. 1838(3): p. 731-8.
  3. Santiago Valtierra, F.X., et al., Sphingomyelins and ceramides with VLCPUFAs are excluded from low-density raft-like domains in differentiating spermatogenic cells. J Lipid Res, 2017. 58(3): p. 529-542.
  4. Aksglaede, L. and A. Juul, Testicular function and fertility in men with Klinefelter syndrome: a review. Eur J Endocrinol, 2013. 168(4): p. R67-76.
  5. Sa, R., et al., The Klinefelter Syndrome and Testicular Sperm Retrieval Outcomes. Genes (Basel), 2023. 14(3).
  6. Umapathy, E., S. Manimekalai, and P. Govindarajulu, Lipid pattern in split ejaculate and Klinefelter's syndrome. Fertil Steril, 1980. 33(3): p. 294-6.

Round 2

Reviewer 3 Report

Comments and Suggestions for Authors

I am truly impressed by this; however, I would appreciate if the authors could clarify and distinguish in few sentences these type of non-lipid rafts from the conventional lipid rafts found in all cell types. If possible, I would encourage the authors to cite, highlighting the appropriate distinctions, a review article published in early 2023 in the Journal of Clinical Medicine that provides an in-depth discussion of recent advances in the molecular mechanisms and signaling pathways associated with lipid rafts.

Author Response

Response to Reviewer 3 – Round 2

Comment 1:

I am truly impressed by this; however, I would appreciate if the authors could clarify and distinguish in a few sentences these type of non-lipid rafts from the conventional lipid rafts found in all cell types. If possible, I would encourage the authors to cite, highlighting the appropriate distinctions, a review article published in early 2023 in the Journal of Clinical Medicine that provides an in-depth discussion of recent advances in the molecular mechanisms and signaling pathways associated with lipid rafts.

Response:

In response to the initial round of comments, we added a paragraph in the Discussion section (lines 257–268) explaining the biophysical characteristics of sphingomyelin and ceramide species containing very long-chain polyunsaturated fatty acids (VLCPUFAs). We emphasized that these species differ markedly from conventional lipid raft-associated lipids, particularly due to their reduced affinity for cholesterol and exclusion from low-density, raft-like membrane fractions in spermatogenic cells. Thus, these lipids do not contribute to classical cholesterol-rich microdomains, often referred to as lipid rafts. This distinction, we believe, satisfies the request for clarification between conventional lipid rafts and the atypical lipid environment described in our study.

Regarding the suggestion to cite a review article published in early 2023 in the Journal of Clinical Medicine, we assume the reference is to the work by Capozzi et al. (2023) titled "Advances in the Pathophysiology of Thrombosis in Antiphospholipid Syndrome: Molecular Mechanisms and Signaling through Lipid Rafts", if this is the case, we respectfully believe that this citation would not be appropriate for our manuscript. While this is an excellent and comprehensive review within the context of immune and vascular disorders, including the role of lipid rafts in antiphospholipid syndrome (APS), its scope is entirely different from the objectives and content of our study.

Our research focuses exclusively on the unique lipid profile of highly motile human spermatozoa obtained through the swim-up method, specifically characterizing sphingomyelin and ceramide species enriched in VLCPUFAs. The molecular and physiological context of sperm function and fertility differs substantially from the immune signaling and pathophysiological mechanisms explored in APS. Therefore, including a citation related to thrombosis in autoimmune disease would introduce a conceptual disconnect for the reader and detract from the central findings of our study.

In summary, we have addressed the reviewer’s initial request by explaining why these atypical sphingolipid species are not associated with lipid rafts. While we greatly appreciate the recommendation to cite additional literature, we hope the reviewer agrees that the specific review in question, though valuable in its own right, falls outside the thematic and biological scope of our manuscript.